# Impact of health promotion strategies on HPV vaccination uptake: A descriptive epidemiological study (2019–2024)

Mirjana Štrbac[1]*, Snežana Ukropina[1], Nataša Nikolić[1,2], Katarina Mašić[3], Smiljana Rajčević[1,2], Dušan Čanković[1,2], Vesna Vuleković[4], Ljerka Popov[4], Mioljub Ristić[1,2]

1 Institute of Public Health of Vojvodina, Novi Sad, Serbia, 2 Faculty of Medicine, University of Novi Sad, Novi Sad, Serbia, 3 Faculty of Philosophy, University of Novi Sad, Novi Sad, Serbia, 4 Health Center Novi Sad, Novi Sad, Serbia

* mirjana.strbac@izjzv.org.rs

## Abstract

### Introduction/objectives

The effectiveness of vaccines depends not only on resource availability but also on broad public acceptance and the uptake of widely accessible vaccines. A vaccination campaign is a strategically coordinated initiative designed to enhance vaccine coverage within a specific population. This study evaluated the impact of health promotion strategies (HPS), including social marketing and education, on human papillomavirus (HPV) vaccine uptake among adolescents aged 9–19 years in City of Novi Sad, Serbia (population ~300,000). Since 2020, efforts have transitioned from individual to organized, publicly funded initiatives.

### Materials and methods

A descriptive epidemiological study was conducted using anonymized data from the electronic immunization registry (2019–2024). Data on HPS implementation were obtained from the Institute of Public Health of Vojvodina. Statistical analyses included Mann–Whitney U test, ANOVA, correlation, and multiple regression with time lag.

### Results and discussion

From 2019 to 2024, 6,395 adolescents received the HPV vaccine, with sharp increases in 2023–2024. Among the different strategies analyzed, health promotion through educational media content delivered by physicians (TV, radio, social networks) and the implementation of the "Open Door" initiative had the most consistent and positive association with increased vaccination coverage. Vaccination uptake was strongly associated with frequent and accessible promotional activities, especially the "Open Door" initiative (r =.668, p <.01; β =.687, p <.001). Media activities

**Citation:** Štrbac M, Ukropina S, Nikolić N, Mašić K, Rajčević S, Čanković D, et al. (2025) Impact of health promotion strategies on HPV vaccination uptake: A descriptive epidemiological study (2019–2024). PLoS One 20(9): e0331592. https://doi.org/10.1371/journal.pone.0331592

**Data availability statement:** All relevant data are within the paper and its Supporting Information files.

**Funding:** The author(s) received no specific funding for this work.

**Competing interests:** No authors have competing interests.

showed moderate effects (r =.270, p <.05). Educational activities show only a weak or non-significant correlation with vaccination rates, except for their modest association with website updates (r =.282, p <.05).

## Conclusions

Accessible, action-oriented interventions, particularly "Open Door" days, were the most effective strategy for increasing adolescent HPV vaccination. Social marketing combining convenience and multi-channel communication significantly enhanced uptake. These findings support the implementation of targeted, barrier-reducing public health strategies to improve vaccine coverage.

## Introduction

The effectiveness of vaccines depends not only on the availability of resources, but also on broad public acceptance and uptake of readily accessible vaccines. Vaccine hesitancy refers to the reluctance or refusal to receive vaccines, even when immunization services are readily accessible [1,2]. During the COVID-19 pandemic, political ideologies have played an increasingly important role in shaping vaccine hesitancy, with studies indicating that political affiliation is a key determinant of vaccine acceptance [3,4]. Strategic health communication serves as a critical tool for mitigating vaccine hesitancy [5–7].

Cervical cancer, primarily caused by human papillomavirus (HPV), remains a major global public health challenge and is the fourth most common cancer among women worldwide. In 2023, an estimated 604,127 new cases and 341,831 deaths were reported globally. Additionally, approximately 12% of adult women with normal cytological findings are infected with HPV, regardless of genotype; Serbia ranks third in Europe in both cervical cancer incidence and mortality, highlighting a pressing public health concern [8].

According to the newly introduced Health Policy paper, HPV immunization is recognized as one of 25 so-called "quick buys" — a set of evidence-based, cost-effective interventions with measurable public health impact within five years [9]. All interventions for cervical cancer have shown immediate effects on the 90-70-90 targets outlined in the Global Strategy to Eliminate Cervical Cancer. These targets include: 90% of girls fully vaccinated with the HPV vaccine by age 15; 70% of women screened with a high-performance test by age 35 and again by age 45; 90% of women with precancerous lesions treated; and 90% of women with invasive cancer receiving appropriate management. Key interventions include HPV vaccination (1–2 doses) for girls aged 9–14 years and HPV DNA screening every 5–10 years starting at age 30 [9,10].

A vaccination campaign is a strategically coordinated initiative aimed at increasing vaccine coverage within a specific population. Such campaigns are typically implemented to rapidly deliver vaccines to large populations, particularly during outbreaks or as part of broader efforts to improve immunization rates. They may be conducted

at national or subnational levels and can target either a single vaccine or multiple vaccines, depending on public health priorities [11].

In Serbia, among a population of 3.82 million women, more than 1,087 new cases of cervical cancer are diagnosed annually, leading to approximately 453 deaths each year [12]. Additionally, genital warts caused by HPV types 6 and 11, are highly prevalent, particularly among adolescents aged 15–19 years in Novi Sad [13]. This epidemiological landscape underscores the urgent need for effective prevention strategies, including HPV vaccination. Given that over 90% of cervical cancer cases are attributable to HPV infection, preventing HPV transmission is a critical measure for reducing the incidence of the disease [14]. HPV vaccination is one of the most effective strategies for preventing cervical cancer, particularly when administered to adolescents before the onset of sexual activity. The vaccine provides 80–100% protection against anogenital warts and reduces the incidence of premalignant lesions by 60–80% among vaccinated individuals [14–16]. The public health impact of HPV vaccination is most pronounced in populations with high vaccine coverage, particularly among women who receive the vaccine prior to HPV exposure [16]. HPV is among the most prevalent viral infections in sexually active individuals. While most HPV infections are asymptomatic and self-limiting, persistent infections can result in severe health complications [17].

The HPV vaccine can be administered as early as age 9, with catch-up vaccination recommended for females up to age 26. In Serbia, since 2008, the HPV vaccine has been recommended for children before their first sexual intercourse and, as of July 2022, has been available free of charge on a gender-neutral basis for adolescents aged 9–19 years [18]. Despite strong evidence supporting the effectiveness of the HPV vaccine in cancer prevention, HPV vaccination programs targeting children have generated considerable public debate, particularly among parents [19]. Common parental concerns include skepticism regarding the vaccine's long-term efficacy and safety, the perception that the recommended vaccination age is too young, distrust of pharmaceutical companies, and fears that vaccination may encourage early sexual activity among children [20]. In Serbia, the first year of the national HPV immunization program (2022/2023 academic year) concluded with a vaccination coverage rate of just around 5% [21].

The HPV vaccination program in Serbia primarily targets young adolescents and requires parental consent prior to vaccine administration. Implemented nationwide since 2022, the program follows a facility-based approach, with vaccines administered by teams consisting of pediatricians and pediatric nurses. Healthcare providers play a crucial role in shaping parental decisions regarding HPV vaccination, as physicians are widely regarded as the most trusted and authoritative sources of information. Their knowledge and attitudes toward the vaccine significantly influence parental choices [22]. As previously noted, HPV immunization is recommended in Serbia, and vaccines have remained consistently available throughout the observed period. However, there is limited published data on the factors influencing Serbian parents' decisions regarding HPV vaccination for their children, particularly since the vaccine became widely available nationwide.

This study aimed to assess the impact of different health promotion strategies (HPS)—social marketing ("promotional activities") and educational measures—on HPV vaccination rates among adolescents aged 9–19 years in the local community of the City of Novi Sad. It also explored temporal trends, potential correlations with promotional efforts, and key predictors of increased vaccine uptake to support future HPV vaccination campaigns in Serbia.

## Background of the HPV vaccination program in Novi Sad, Serbia

Novi Sad, a city in northern Serbia, has a population of approximately 350,000 and is the second largest city in the country. Each age cohort includes around 4,000 children, with an even gender distribution (approximately 50% girls and 50% boys). In total, about 44,000 children aged 9–19 years reside in Novi Sad. Vaccination — both mandatory and recommended—is provided through the primary healthcare system by designated pediatricians at the Novi Sad Health Center [21,22].

In 2019, the HPV vaccination was administered sporadically, largely depending on parents' willingness to purchase the vaccine and coordinate its administration with pediatricians. In 2020, however, Novi Sad became the first city in Serbia

to launch a promotional campaign for HPV immunization, offering the quadrivalent HPV vaccine free of charge. This campaign primarily targeted girls aged 12–18 years, marking a significant step toward increasing vaccination coverage. In 2021, eligibility for free vaccination was expanded to include boys aged 12–18 years, further strengthening the vaccination efforts. The initiative gained momentum with the national introduction of HPV immunization in Serbia in July 2022. At that point, the nonavalent HPV vaccine was made available in Novi Sad at no cost to all adolescents aged 9–19 years, ensuring broader access for both genders. Despite the challenges presented by the COVID-19 pandemic and the growing of anti-vaccination sentiments since 2020, the Institute of Public Health of Vojvodina (IPHV) has remained committed to promoting HPV vaccination. These efforts continue to emphasize immunization as a crucial preventive measure in the fight against cervical cancer [21,22].

## Materials and methods

A descriptive epidemiological study was conducted using fully anonymized data from electronic immunization registry at the Primary Health Care Center located in Novi Sad. Authors had no access to information that could identify individual participants during or after data collection. The study analysed anonymized immunization records from the HPV immunization registry, including the birth dates, gender, and the administration of the first dose of the HPV vaccine for all children vaccinated between January 1, 2019, and December 31, 2024. Data were accessed for research purposes on January 13, 2025. Data about the participants' parents were not included. HPS implemented by healthcare centers across South Bačka District, as well as in the city of Novi Sad, were documented and described by the Health Promotion Department at the IPHV.

The study protocol was approved by the Ethics Committee of the Primary Health Care Center "Novi Sad", City of Novi Sad (approval number: 21/20–1, dated October 21, 2024).

## Health-promotion strategies until december 2024

HPS comprised eight key activity groups–five social marketing initiatives and three educational activities – designed to address knowledge gaps regarding HPV vaccines, attitudes toward HPV immunization hesitancy, and to increase the number of individuals vaccinated against HPV in City of Novi Sad. This approach was based on a prior systematic review [23]. The HPS aimed at increasing awareness and uptake of HPV immunization included a range of activities, such as organizing walk-in vaccination sessions ("Open Door" events), expert appearances on television, engagement through social media, podcast recordings, informational sessions for parents, printing and distribution of promotional materials across the city, school-based lectures for older adolescents (ages 15–18), and accredited educational sessions for healthcare professionals. The HPS developed and implemented in this study were systematically categorized into two thematic groups, as detailed below.

1. **Social marketing approach ("promotional activities")**

◦ The "Open Door" social marketing campaign for HPV immunization: Emails and Viber messages were sent to parents using address databases from primary and secondary schools, including links to 10 informative articles on the official website of the IPHV. Each "Open Door" event included the completion of detailed medical documentation for the administered HPV vaccines. These records were compiled and submitted to the IPHV as part of the routine immunization reporting system.

◦ Media activities: Experts from IPHV took part in media appearances, while press releases were issued announcing the schedule of "Open Door" session for HPV immunization. Media coverage was monitored through weekly press-clipping analysis, which involved identifying and categorizing relevant articles and broadcasts related to HPV immunization.

- Websites updates: A dedicated webpage was launched on the IPHV website, providing comprehensive information on recommended HPV immunization, including 15 frequently asked questions with corresponding answers. Digital versions of 10 health education materials —such as informative articles, publications, and interactive content— were uploaded, covering topics related to HPV prevention, vaccination, and the prevention of sexually transmitted diseases. These materials included three leaflets, two flyers, one brochure, a PowerPoint presentation, a video clip, and a podcast.

- Social media: Visual content promoting the "Open Door" immunization schedule was shared on Instagram and TikTok to engage both parents and younger audiences. In addition, podcasts related to HPV immunization were hosted on the IPHV YouTube channel.

Website traffic and social media engagement data were collected using analytics tools employed by the IPHV, which tracked the number and type of posts, views, user interactions, and overall reach.

- Distribution of Health Education and Promotion Materials: Informational posters were displayed in public transportation vehicles, including both urban and suburban buses. A total of 2,500 leaflets on HPV infection —targeting both female and male adolescents— and 5,000 leaflets on HPV immunization were distributed. Additionally, QR codes were generated to provide quick access to the digital health education materials, supporting pediatricians in individual counseling sessions on HPV immunization. Data on dissemination were collected using signed distribution lists, specifying the title and print run of each distributed item.

2. **Face-to-face education for students, parents, and healthcare professionals**

- Lectures and interactive discussions were organized in high schools and student dormitories to engage students in discussions about HPV prevention and immunization. Health-education activities for parents were also conducted separately.

- Lectures and interactive discussions were held with parent representatives from both primary and secondary schools to inform and educate them about the importance of HPV vaccination. Educational sessions for young people and parents included a peer-sharing component. Parents were encouraged to share digital versions of presentations and brochures with other parents and students in their child's class.

- Accredited professional meetings were organized for healthcare professionals—including pediatricians, general practitioners, epidemiologists, pharmacists, gynaecologists, nurses, and medical technicians — with the aim of enhancing their knowledge of HPV prevention and immunization. Educational sessions for healthcare professionals were evaluated through the standardized methodology for accredited education within the Serbian healthcare system. Participants received individual certificates, and the implementation was officially reported to the Health Council of Serbia.

**Design and implementation of campaigns to improve HPV vaccine uptake**

In March 2024, a new approach to promotional activities was introduced under the initiative called "Open Door" sessions, which enabled children to receive HPV vaccination in pediatric clinics without prior appointments. The goal was to improve access to healthcare services and increase vaccination uptake. Clear and direct messages containing information on vaccination locations, dates, and times were disseminated via various media channels, emails, and Viber messages. The IPHV, with approval from the management of the Primary Health Care Center Novi Sad, contacted school principals across the City of Novi Sad by phone and email. Principals shared the information with class teachers, who, as part of routine school administrative procedures, maintain parents' contact details (telephone numbers). These existing records were used to send SMS or Viber messages to parents. Information was also forwarded to school parent councils to ensure broader coverage. Participation in the vaccination campaign was entirely voluntary, and the decision to bring a

child for vaccination was left to the parents. No additional consent was required, as this communication fell within standard school–parent information practices. In addition, The "Open Door" immunization sessions were launched in March 2024 and continued through December 2024. Communication with parents via emails and Viber messages was conducted regularly during this period, in alignment with the vaccination schedule. Media activities, including expert TV appearances and press releases, began in March 2024. Updates to the dedicated HPV website and the launch of social media campaigns on Instagram, TikTok, and YouTube started in April 2024 and were actively maintained until the end of the campaign. Distribution of printed materials began in May 2024. Educational lectures and workshops for students and parents were held primarily during the second and third quarters of 2024. Accredited continuing medical education sessions for healthcare providers were organized throughout the campaign, starting in May 2024.

### Statistical analysis

The collected data were processed using the IBM SPSS Statistics 26 software package. In line with the objectives of this study, the following data analysis methods were applied: descriptive statistics, one-way analysis of variance, multiple regression with lag, correlation analysis, and Mann-Whitney U test. Normality of the monthly vaccination data distribution was assessed using the Shapiro–Wilk test. The results indicated that data for most years did not meet the assumption of normality (e.g., $p < .05$ for 2020, 2021, 2023, and 2024), thereby justifying the use of nonparametric methods for comparisons between years. Before conducting ANOVA, the assumptions of normality and homogeneity of variances were assessed using the Shapiro–Wilk and Levene's tests, respectively. The results indicated deviations from normality ($p < .05$) in some groups (e.g., boys and girls), and Levene's test revealed unequal variances between them. Despite these violations, ANOVA was retained for exploratory comparisons due to equal group sizes and the value of planned post hoc analyses. To address these limitations, nonparametric alternatives (e.g., Mann–Whitney U test) were also applied, and results were interpreted with caution. For regression analyses, diagnostic checks supported the assumptions of normality, linearity, homoscedasticity, and absence of multicollinearity. The P–P plot of standardized residuals suggested that residuals were approximately normally distributed. Visual inspection of residuals confirmed linearity and equal variance. Variance Inflation Factor (VIF) values ranged from 1.13 to 2.30, indicating that multicollinearity was not a concern.

### Results

Between 2019 and 2024, a total of 6,395 individuals aged 9–19 years received the HPV vaccine. The data presented indicate a clear upward trend in the number of vaccinated individuals over the years, with a marked acceleration in the most recent period (2022–2024). Notably, the year 2024 shows a significant range (428) (Fig 1).

Table 1 shows a clear increase in the number of vaccinated individuals from 2019 to 2024. In 2019, the average number of vaccinated individuals was 1.67, with minimal variability. By 2020, the average increase to 30.67, altough variability also rose, as indicated by a high standard deviation of 57.55. The upward trend continued in 2021 (50.33) and 2022 (125.33), with variability increasing each year. In 2023, the average reached 140.92, with a slightly lower standard deviation of 47.38. In 2024, the average further increased to 188.83, accompanied by the highest observed variability, as reflected by a standard deviation of 136.57.

Table 2 presents statistical comparisons of HPV vaccination rates between different years (2019–2024) using the Mann-Whitney U test. The results show statistically significant differences between each pair of years, with all p-values reported as <.001. As the years progress, both the U-values and the mean ranks for the latter year in each comparison consistently increase, suggesting a steady upward trend in the number of vaccinated individuals. For instance, the mean rank for 2020 is 203.50, whereas by 2024 it reaches 2,796.50, reflecting the continuous growth in vaccination uptake over time.

More specifically, Fig 2 illustrates the monthly number of HPV-vaccinated individuals aged 9–19 years from 2019 to 2024. A clear upward trend is observed over the years, with significant peaks in 2023 and 2024. The most prominent

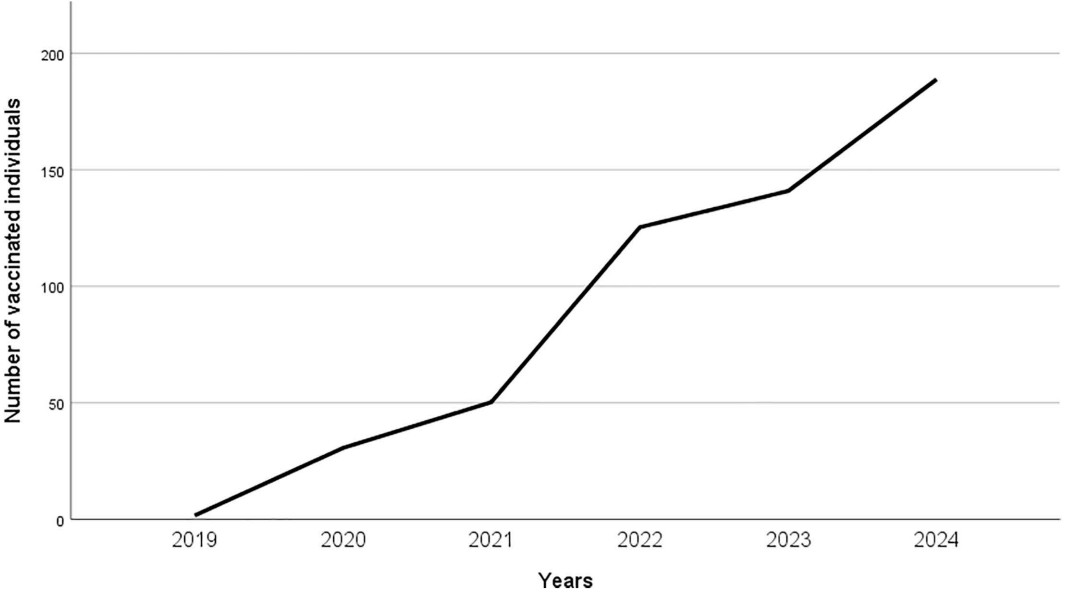

**Fig 1. Number of vaccinated individuals aged 9–19 years (2019-2024).**

**Table 1. Annual number of vaccinated individuals aged 9-19 years by year (2019–2024).**

| Year | M [a] | SD [b] | Min [c] | Max [d] |
|------|-------|--------|---------|---------|
| 2019 | 1.67 | 1.15 | 0 | 4 |
| 2020 | 30.67 | 57.55 | 0 | 165 |
| 2021 | 50.33 | 60.06 | 3 | 170 |
| 2022 | 125.33 | 62.24 | 38 | 242 |
| 2023 | 140.92 | 47.38 | 89 | 267 |
| 2024 | 188.83 | 136.57 | 62 | 490 |

[a] Mean; [b] Standard deviation; [c] Mimimum; [d] Maximum.

surge occurred in March 2024, reaching nearly 500 vaccinations, followed by another peak in October of the same year. In 2023, a steady increase in vaccination uptake was also observed, particularly during the first quarter. In contrast, the period 2019–2021 was characterized by relatively low and stable vaccination numbers.

Fig 3 illustrates the monthly number of HPV-vaccinated individuals aged 9–19 years (blue bars) alongside the frequency of promotional campaigns (red line) from 2019 to 2024. A clear upward trend in vaccination uptake is observed over time, with noticeable fluctuations that correspond to peaks in promotional activities. Although the peaks in promotional campaigns do not always align precisely with increases in the number of vaccinated individuals, the most substantial rises in vaccination numbers coincide with periods of intensified campaign efforts — particularly in 2023 and 2024 —suggesting a strong association between promotional activities and vaccination uptake.

Further analysis demonstrates that "Open Door" vaccination initiatives exibit the strongest positive correlation with the number of vaccinated individuals, particularly among males ($r = .668$, $p < .01$) and children overall ($r = .588$, $p < .01$). This suggests that reducing logistical barriers and providing walk-in vaccination opportunities significantly enhance vaccine uptake. Media activity also shows statistically significant correlations with vaccination rates for females ($r = .263$, $p < .05$), males ($r = .250$, $p < .05$), and children ($r = .270$, $p < .05$), highlighting the role of public communication in increasing

**Table 2. Comparisons of HPV vaccination rates between years (2019–2024).**

| Years | U | Z | p | Mean First Year Rank | Mean Second Year Rank |
|---|---|---|---|---|---|
| 2019 vs 2020 | 0 | −19.621 | <.001 | 10.50 | 203.50 |
| 2019 vs 2021 | 0 | −24.900 | <.001 | 10.50 | 321.00 |
| 2019 vs 2022 | 0 | −38.884 | <.001 | 10.50 | 767.00 |
| 2019 vs 2023 | 0 | −41.183 | <.001 | 10.50 | 859.00 |
| 2019 vs 2024 | 0 | −47.508 | <.001 | 10.50 | 1139.50 |
| 2020 vs 2021 | 0 | −31.081 | <.001 | 183.50 | 667.00 |
| 2020 vs 2022 | 0 | −43.105 | <.001 | 183.50 | 1113.00 |
| 2020 vs 2023 | 0 | −45.188 | <.001 | 183.50 | 1205.00 |
| 2020 vs 2024 | 0 | −51.020 | <.001 | 183.50 | 1485.50 |
| 2021 vs 2022 | 0 | −45.749 | <.001 | 301.00 | 1348.00 |
| 2021 vs 2023 | 0 | −47.718 | <.001 | 301.00 | 1440.00 |
| 2021 vs 2024 | 0 | −53.273 | <.001 | 301.00 | 1720.50 |
| 2022 vs 2023 | 0 | −56.294 | <.001 | 747.00 | 2332.00 |
| 2022 vs 2024 | 0 | −61.074 | <.001 | 747.00 | 2612.50 |
| 2023 vs 2024 | 0 | −62.562 | <.001 | 839.00 | 2796.50 |

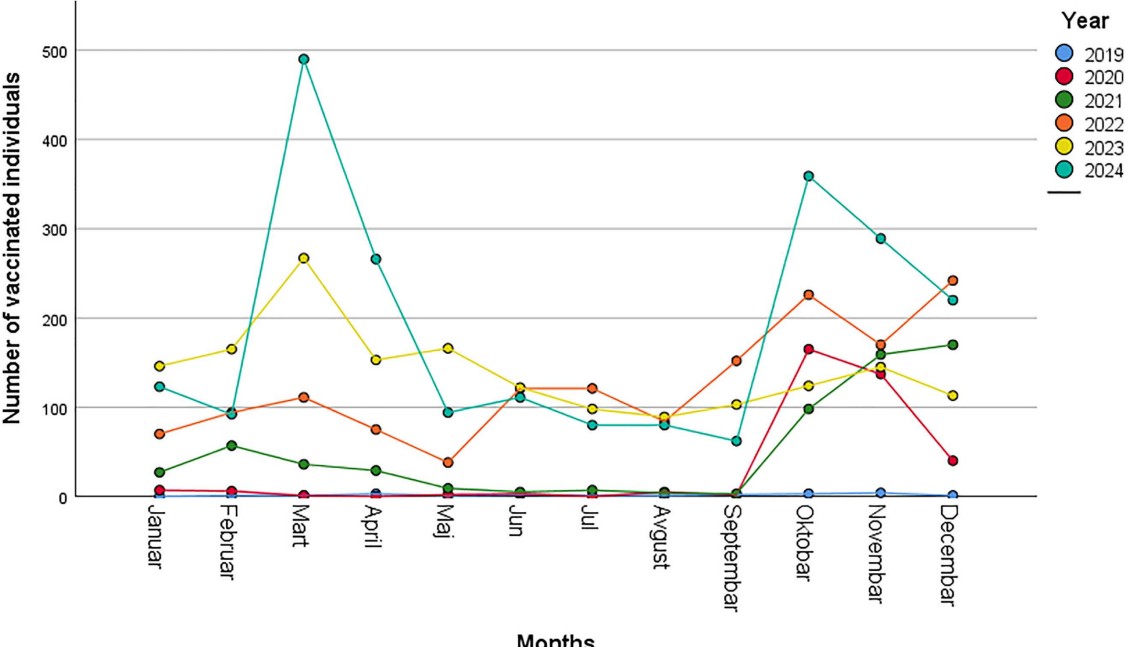

**Fig 2. Monthly trends of HPV vaccination among individuals aged 9–19 years (2019–2024).** Legend: The green line representing the number of vaccinated individuals aged 9–19 years shows the highest peaks in March and October 2024, the months during which the "Open Door" campaign was implemented most intensively.

awareness and participation. Furthermore, website updates are positively associated with vaccination numbers for males (r = .241, p < .05) and children (r = .241, p < .05), suggesting that digital information sources contribute to engagement. A strong interrelationship is observed between media activity and social media presence (r = .618, p < .01), as well as

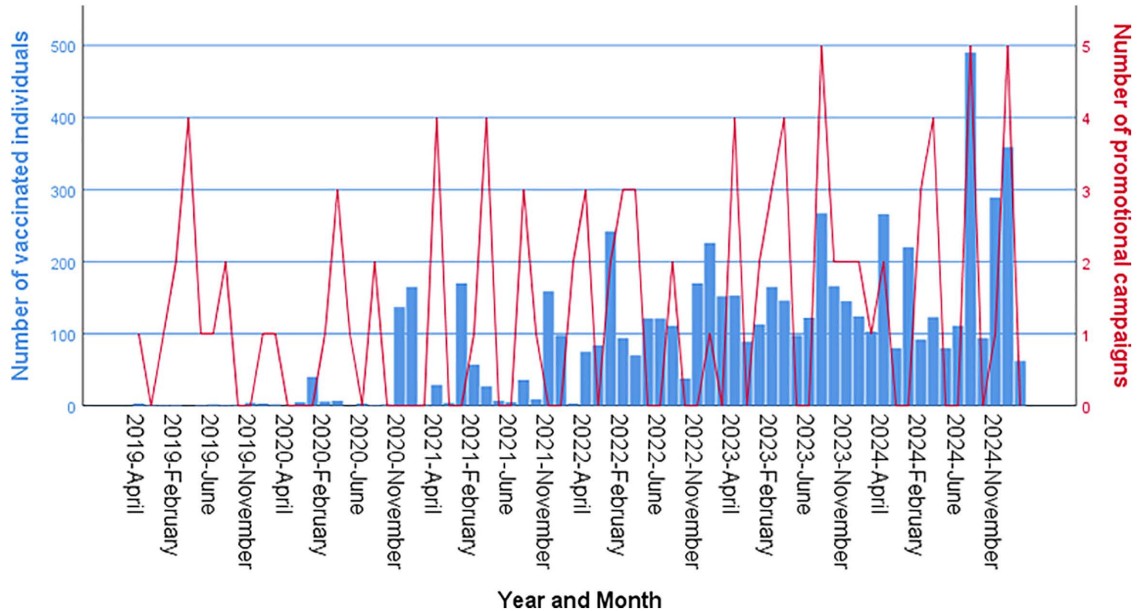

**Fig 3. Promotional activities and HPV vaccination uptake (2019–2024).** Legend: The red line in the graph represents the number of promotional campaigns, which does not consistently correlate with the number of vaccinated individuals (blue bars) from 2019 to 2024. *However, in 2024, when the "Open Door" campaign was implemented, both the number of promotional campaigns and the number of vaccinated individuals reached their highest levels.*

between website updates and social media activity (r = .615, p < .01), emphasizing the importance of integrated communication strategies. Similarly, the distribution of promotional materials correlates with social media activity (r = .438, p < .01) and website updates (r = .434, p < .01), reinforcing the effectiveness of multi-channel outreach. Interestingly, educational activities show only weak or non-significant correlation with vaccination rates, except for their modest association with website updates (r = .282, p < .05). This suggests that while educational efforts may increase awareness, they are less directly linked to vaccine uptake compared to more accessible, action-oriented interventions such as the "Open Door" vaccination days (Table 3).

Additionally, a multiple regression analysis with a one-month time lag was performed, incorporated lagged values of "promotional activities" as predictors to assess whether activities from the previous months influenced the number of vaccinated individuals in subsequent period. This time-lagged model explained a higher proportion of variance in vaccination uptake ($R^2$ = .555, F(12, 58) = 6.024, p < .001). **"Open Door"** events remained the most significant predictor, demonstrating both immediate and delayed effects ($\beta$ = .354, t = 3.403, p = .001). The distribution of promotional materials approached statistical significance (p = .054), and showed a delayed negative effect (with a delay) ($\beta$ = −.209, t = −1.970, p = .054). In contrast, educational activities did not exhibit a statistically significant delayed effect ($\beta$ = .170, t = 1.759, p = .084).

Analysis of the distribution by gender (Fig 4) reveals a significantly higher vaccination uptake among females, with 4,714 (73.7%) individuals vaccinated compared to 1,681 males (26.3%). The highest number of vaccinated individuals was recorded in the 12–15 age group, peaking at age 14, when 1,100 individuals (17.2% of the total vaccinated population) received the vaccine. Among males, the highest uptake was observed at age 15, with 345 individuals vaccinated (20.5% of all vaccinated males), whereas among females, the peak occurred at age 14, with 782 individuals vaccinated (16.6% of all vaccinated females). After age 15, vaccination rates progressively declined, with a more pronounced decrease after age 16, suggesting lower vaccine acceptance among older adolescents.

In addition to age-related trends, the data highlight a substantial gender gap, particularly in the 10–13 age group.

**Table 3. Correlation between health promotion strategies and HPV vaccination uptake.**

| Vaccination uptake | Health promotion strategies | | | | | |
|---|---|---|---|---|---|---|
| | Educational activities | Open Door activities | Media activity | Social media | Website updates | Distribution of materials |
| Vaccinated females | .123 | .506 [b] | .263 [a] | .193 | .224 | .092 |
| Vaccinated males | .194 | .668 [b] | .250 [a] | .229 | .241 [a] | .174 |
| Vaccinated children (in total) | .155 | .588 [b] | .270 [a] | .215 | .241 [a] | .126 |

[a]$p < 0.05$; [b] $p < 0.01$.

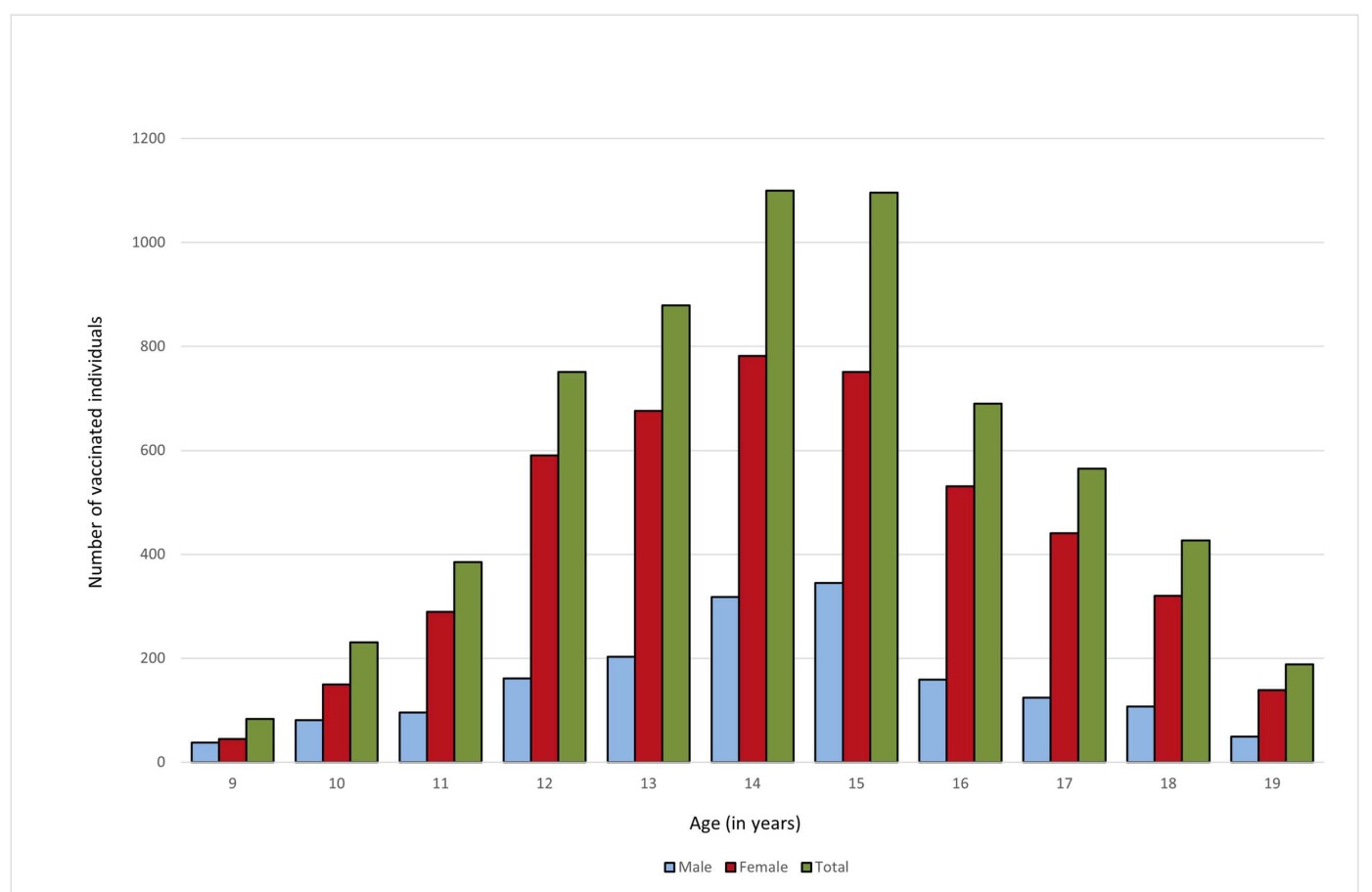

**Fig 4. Number of vaccinated individuals by age and gender (2019-2024).**

An analysis of variance (ANOVA) showed a statistically significant difference in the number of vaccinated boys across vaccination years (2019–2024) ($F_{(5, 66)} = 16.991$, $p < .001$), with the highest numbers recorded in 2024. Post hoc Tukey HSD tests (S1 Table) confirmed significant year-to-year differences, particularly a sharp increase in 2023 and a peak in 2024. A multiple regression analysis identified the "Open Door" initiative ($\beta = 0.697$, $p < .001$) and educational activities ($\beta = 0.240$, $p < .05$) as the strongest predictors of vaccination rates among boys ($R^2 = 0.517$, $F_{(6, 65)} = 11.588$, $p < .001$).

ANOVA also revealed significant differences in the number of vaccinated girls across vaccination years ($F_{(5, 66)}$ = 9.200, $p < .001$), indicating a steady increase over time. Post hoc Tukey HSD tests (S2 Table) showed that 2022, 2023, and 2024 differed significantly from 2019 and 2020, while no significant differences were found between 2022–2023 or 2023–2024. Multiple regression analysis confirmed the "Open Door" initiative as the only significant predictor of vaccination uptake among girls ($β = 0.511$, $t = 4.512$, $p < .001$; $R^2 = 0.313$, $F_{(6, 65)}$ = 4.932, $p < .001$).

In addition, two multiple regression analyses were conducted to assess the impact of promotional activities on HPV vaccination uptake among individuals aged 9–14 and 15–19 years. For the 9–14 age group, the model was significant ($R^2 = 0.470$, $F_{(6, 65)}$ = 9.620, $p < .001$), with the "Open Door" initiative ($β = 0.687$, $p < .001$) and educational activities ($β = 0.202$, $p < .05$) emerging as significant predictors. For the 15–19 age group, the model was also significant ($R^2 = 0.302$, $F_{(6, 65)}$ = 4.696, $p = .001$), with the "Open Door" initiative as the sole significant predictor ($β = 0.440$, $p < .001$).

## Discussion

This study demonstrated a steady and substantial increase in HPV vaccination uptake among adolescents aged 9–19 years in City of Novi Sad between 2019 and 2024, with the most pronounced rise observed from 2022 onward. Although girls consistently accounted for the majority of vaccinated individuals, a notable increase was also recorded among boys, particularly in the last two observed years. The highest uptake was observed among adolescents aged 12–15 years, with a clear decline in older age groups. The findings suggest that health promotion activities—especially the "Open Door" initiative—played a pivotal role in increasing vaccine uptake, not only through immediate effects but also with delayed impact over time. In contrast, educational interventions showed weaker associations. Strong correlations between different communication channels (e.g., media, social media, websites) underscore the importance of integrated outreach strategies. Overall, the results highlight the combined importance of accessibility and targeted communication in improving HPV vaccination coverage. It is important to note that while some promotional messages and materials may have reached audiences beyond City of Novi Sad—within the South Bačka District (in which Novi Sad is located) and the Autonomous Province of Vojvodina (for which Novi Sad serves as the main administrative center)—via media and digital platforms, the core campaign activities were primarily focused on the City of Novi Sad. Most importantly, both the monitoring and evaluation of HPV vaccination uptake were strictly limited to the population of the City of Novi Sad, as vaccination data were collected exclusively for this city.

Despite a clear year-over-year increase, HPV vaccination coverage in the observed area remains suboptimal. Efforts should focus on overcoming cultural barriers, dispelling myths, addressing healthcare worker hesitancy, and strengthening trust through routine care and proactive communication [22,24]. Communication strategies must be introduced prior to vaccine rollout [25], while ongoing evaluation is essential to identify effective, context-specific interventions. Our findings emphasize the need to reduce access barriers, which should be a central goal of future policies to improve coverage.

A recent systematic review and meta-analysis from sub-Saharan Africa showed that communication strategies substantially improve HPV vaccine uptake among adolescents. Decision-facilitating interventions (e.g., decision aids, motivational interviewing) achieved the highest uptake (up to 100%), followed by enabling (92%) and informational approaches (90%). Targeting healthcare workers, community leaders, and school staff was particularly effective (uptake ≥92%). Interactive formats such as training and community drama outperformed traditional information, education and communication materials. These findings highlight the value of tailored, multilevel communication strategies to support HPV vaccination in low-resource setting [26].

Another systematic review demonstrated that such interventions were implemented through various methods, including electronic health record prompts (e.g., recall/reminder systems), text messaging, automated phone calls, interactive computer videos, and email. Technology-based approaches were associated with higher rates of vaccine initiation and completion compared to control groups [23], findings that align with our own results. In our study, communication-based activities—particularly those focused on education and explaining the benefits of vaccination—played a significant,

though not the most influential, role in increasing the number of vaccinated boys and girls in Novi Sad. Although parents of adolescents were the primary target group, the campaign also actively engaged adolescents through school-based educational sessions, as well as healthcare providers through accredited professional education. Furthermore, the general public was reached via diverse media channels, digital content, and the distribution of promotional materials in public spaces, ensuring a comprehensive, multi-audience approach.

As noted in previous studies, including Cartmell KB et al., effective HPV vaccination messaging should emphasize cancer prevention over sexual transmission, promote routine vaccination, and highlight HPV-related risks and costs. Tailoring messages to specific demographic groups and delivering them across diverse media platforms was recommended to ensure consistent, accurate information [27]. Appeals to parental responsibility, the high prevalence of HPV, and evidence of non-sexual transmission were also effective. Trusted messengers—such as peers, healthcare providers, and public health institutions—played a key role. An Austrian study found parental knowledge strongly associated with vaccine acceptance for daughters, but not sons [28]. Similarly, research from China advised campaigns to focus on parents of girls aged 13–15 years, using social media to convey clear, evidence-based messages about HPV vaccine benefits [29].

In our study, aside from expert commentary, we did not implement more intensive television-based promotional activities. This approach may warrant consideration in future interventions, as previous research has identified it as highly effective [30]. Moreover, our communication strategy did not include the use of reminder systems. However, evidence from a stepped-wedge cluster randomized trial has shown that reminder-recall letters effectively prompted the majority of parents to complete the HPV vaccination series. Despite this, persistent vaccine-related misconceptions remain a barrier, highlighting the need for complementary and targeted communication strategies [31].

Findings from a qualitative study aimed at gathering user feedback and identifying the strengths and limitations of educational materials designed to promote HPV immunization suggest that such resources may be well-received and effective in enhancing HPV-related knowledge, increasing confidence in vaccine safety and efficacy, and strengthening vaccination intent [32]. Although our study did not employ qualitative methods, it represents one of the few investigations in Serbia to establish a link between health promotion strategies and measurable outcomes—specifically, increased HPV vaccination coverage.

A systematic review and meta-analysis by Oh NL et al. identified provider communication (PC) as a key driver of HPV vaccine uptake in the U.S., showing increased initiation among patients after PC, across genders, age groups, and over time. Provider discussions alone had an effect comparable to overall PC [33]. In our study, health promotion strategies were developed by public health specialists, while PC was primarily conducted by public health experts, pediatricians, and general practitioners.

It should be noted that, although our study examined an HPV vaccination campaign conducted during the COVID-19 pandemic, its success was not compromised—similar to the findings of a recently published study from Tennessee [34].

Globally, barriers to MMR and HPV vaccination include healthcare delivery [35], challenges such as limited time for vaccine discussions, privacy concerns among minority groups, language barriers, and difficulties navigating the healthcare system [26,35,36]. Other obstacles involve geographic distance, restricted clinic hours, and lack of preventive care centers offering HPV vaccines [26,36]. In our study, healthcare service availability, scheduling flexibility, and clear targeted communication with parents were key predictors of increased vaccination rates. Our results indicate that, today, parents place a highly value on easy access to healthcare services, a factor that we successfully addressed through the "Open Door" initiative. Although vaccination coverage increased across all observed years, the most pronounced acceleration coincided with the implementation of this initiative. While causality cannot be directly established due to the observational nature of the study, the strength and consistency of the effect, as demonstrated by the regression analysis, support the relevance of this approach. Other promotional activities included in the model did not emerge as significant predictors, which further reinforces the distinct contribution of the "Open Door" strategy beyond prior awareness efforts or cumulative exposure.

The most recent findings from a study employing a multi-method approach, which involved data collection via open-ended questionnaires distributed to European youth organizations, concluded that the primary weaknesses include insufficient knowledge about HPV and its consequences among adolescents aged 9–19 years, young adults, healthcare professionals, and parents, challenges in accessing vaccination sites, and discomfort in discussing sexually transmitted infections with children. However, these issues can be mitigated through effective educational methods, such as engaging campaign tools and targeted educational meetings [37].

To enhance effectiveness, targeted initiatives should be carefully tailored to the specific needs of each district and designed with consideration for local decision-makers within the community. In our study area, we observed a significantly higher number of vaccinated boys in 2024 compared to all previous years (2019–2023). While promotional efforts seem to influence vaccine uptake, a direct correlation was not consistently evident—certain months show increased activity without a corresponding rise in vaccination rates. Through various educational initiatives in Novi Sad, we constantly emphasize the importance of receiving the vaccine before the first sexual experience. However, it has been noted that parents most frequently bring their children for vaccination at the ages of 14 and 15.

Overall, the study's strengths lie in its extensive timeframe, detailed vaccination and promotional activity data, advanced statistical approach, and focus on real-world, programmatic HPV vaccination efforts in an urban Serbian setting. These aspects enhance the validity and applicability of the findings to inform future vaccination promotion strategies.

This study has several limitations. First, it relied on secondary, anonymized data from the electronic immunization registry, which did not include individual-level sociodemographic characteristics or reasons for HPV vaccine acceptance or refusal. Second, although the implementation of HPS was comprehensively documented, the study was not designed to assess their causal impact on vaccination uptake. Therefore, the temporal associations between HPS and vaccination trends should be interpreted with caution. Third, the influence of potential confounding factors—such as concurrent public health campaigns, broader national immunization efforts, media coverage unrelated to HPS, or school-based health initiatives—could not be excluded and may have affected HPV vaccination rates independently of the interventions described. Fourth, the study may be subject to selection bias. Parents who responded to promotional messages or allowed their children to be vaccinated may have differed systematically from those who did not, potentially leading to an overestimation of the effectiveness of HPS. Additionally, data on vaccine completion (i.e., receipt of the second dose) were not analyzed, limiting the assessment of full immunization coverage. Fifth, our study focused on the collection and analysis of depersonalized data of vaccinated individuals aged 9–19 years, without additional insights into the sociodemographic characteristics of the parents of the vaccinated children. Sixth, the study did not include data on parents' access to digital communication channels (e.g., cell phones, websites, podcasts) or their digital literacy levels, which may have influenced their exposure to certain health-promotion messages. However, to address this potential limitation and ensure broader accessibility, the communication strategy also incorporated non-digital approaches, such as printed educational materials (leaflets, brochures, posters) and in-person educational sessions. This multi-modal strategy aimed to reach a wider audience regardless of their access to or familiarity with digital tools. Lastly, as the study was conducted in a single urban setting, the generalizability of the findings to other regions with different demographic, cultural, or healthcare system characteristics may be limited.

In conclusion, our data indicate a general upward trend in the number of vaccinated individuals—both girls and, notably, boys—over the observed period, with noticeable monthly fluctuations. The findings also confirm that peaks in promotional activities do not always directly correspond to increases in vaccination rates, highlighting the complexity of factors influencing immunization uptake. Importantly, the "Open Door" initiative emerges as a novel and practical approach that effectively reduces logistical barriers and enhances vaccine accessibility. Given its demonstrated success, this strategy holds significant potential for scale-up and adaptation in similar urban and resource-limited settings to improve HPV vaccination coverage.

Building on these findings, future initiatives should focus on rigorous evaluation of long-term vaccine completion rates and cost-effectiveness, as well as on expanding community engagement and tailored communication strategies. Policy-makers are encouraged to incorporate such evidence-based, accessible vaccination models into broader immunization programs to maximize public health impact.

## Supporting information

**S1 Table. Post hoc pairwise comparisons of the number of vaccinated boys by year of vaccination (2019–2024).**
(DOCX)

**S2 Table. Post hoc pairwise comparisons of the number of vaccinated girls by year of vaccination (2019–2024).**
(DOCX)

**S1 File. All vacccinated by years.**
(XLSX)

**S2 File. Vaccinated along with campaigns.**
(XLSX)

## Acknowledgments

The authors would like to thank all colleagues at the Health Center of Novi Sad for their administrative and technical support. Special thanks to Biljana Petrović, Emilija Samardžić and Ankica Vukas for their valuable contributions.

## Author contributions

**Data curation:** Nataša Nikolić, Katarina Mašić, Vesna Vuleković, Ljerka Popov.

**Formal analysis:** Nataša Nikolić, Katarina Mašić.

**Investigation:** Nataša Nikolić, Katarina Mašić.

**Methodology:** Mirjana Štrbac, Mioljub Ristić.

**Project administration:** Mioljub Ristić.

**Resources:** Snežana Ukropina.

**Supervision:** Mirjana Štrbac, Dušan Čanković, Mioljub Ristić.

**Validation:** Smiljana Rajčević, Mioljub Ristić.

**Visualization:** Snežana Ukropina, Katarina Mašić, Smiljana Rajčević.

**Writing – original draft:** Mirjana Štrbac, Snežana Ukropina.

**Writing – review & editing:** Mirjana Štrbac, Mioljub Ristić.

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
