## [Decision Letter · Decision Letter 0]

3 Jul 2025

Dear Dr. Štrbac,

Please submit your revised manuscript by Aug 17 2025 11:59PM. If you will need more time than this to complete your revisions, please reply to this message or contact the journal office at plosone@plos.org . A rebuttal letter that responds to each point raised by the academic editor and reviewer(s). You should upload this letter as a separate file labeled 'Response to Reviewers'.A marked-up copy of your manuscript that highlights changes made to the original version. You should upload this as a separate file labeled 'Revised Manuscript with Track Changes'.An unmarked version of your revised paper without tracked changes. You should upload this as a separate file labeled 'Manuscript'.

We look forward to receiving your revised manuscript.

Kind regards,

Iskra Alexandra Nola

Academic Editor

PLOS ONE

Journal Requirements:

2. We note that your Data Availability Statement is currently as follows: [All relevant data are within the manuscript and its Supporting Information files]

3. We notice that your supplementary tables are included in the manuscript file. Please remove them and upload them with the file type 'Supporting Information'. Please ensure that each Supporting Information file has a legend listed in the manuscript after the references list.

4. We are unable to open your Supporting Information file [data.zip]. Please kindly revise as necessary and re-upload.

Reviewers' comments:

Reviewer's Responses to Questions

**Comments to the Author**

1. Is the manuscript technically sound, and do the data support the conclusions?

Reviewer #1: Yes

Reviewer #2: Partly

2. Has the statistical analysis been performed appropriately and rigorously?

Reviewer #1: Yes

Reviewer #2: Yes

3. Have the authors made all data underlying the findings in their manuscript fully available?

Reviewer #1: Yes

Reviewer #2: Yes

4. Is the manuscript presented in an intelligible fashion and written in standard English?

Reviewer #1: Yes

Reviewer #2: Yes

Reviewer #1: Abstract

• The sentence “...showed moderate effects. Educational efforts had limited influence.” would benefit from numerical clarity (e.g., correlation coefficients or p-values) to support the claims.

• Replace “among adolescents” with “among adolescents aged 9–19 years” for consistency with the main text.

Introduction

• Clarify the sentence “...regardless of type Serbia ranks third…” by correcting the punctuation: insert a period or semicolon between “type” and “Serbia.”

• Provide a citation for the “Health Policy paper” that defines HPV immunization as a “quick buy.”

• The aim statement is too long and somewhat repetitive. It could be tightened to a single paragraph focusing on the core research questions.

• Use consistent terminology (e.g., "nanovalent HPV vaccine" vs. "nonavalent HPV vaccine").

• Clearly define whether the campaign’s reach extended beyond Novi Sad.

Materials and Methods

• Clarify what “address databases” were used—school-registered parent contacts? Was consent implied or granted?

• Include the time frame of implementation for each HPS (e.g., when were TV campaigns launched?).

• Elaborate on the regression model's specifications—how was the time lag introduced and validated?

• Explain rationale for selecting the Mann–Whitney U test over t-tests or nonparametric alternatives.

• Include a brief description of assumptions checked before using ANOVA and regression.

Results

• Table 2 lists all p-values as “.000”—replace with “< .001.”

• For correlation analysis (Table 3), consider grouping HPS by type (social marketing vs. educational) for clarity.

• Some figure captions (e.g., Fig 2, Fig 3) should describe what the reader should take away from the figure (e.g., “note peaks coinciding with Open Door campaigns”).

Discussion

• The first paragraph should more clearly summarize key findings before elaborating on implications.

• Avoid vague statements like “a substantial body of research”; specify examples or summarize meta-analysis findings concisely.

• Discuss the potential impact of confounding factors—e.g., were other health campaigns running concurrently?

Conclusion

• Consider highlighting the novelty of the “Open Door” approach and its potential for scale-up in similar settings.

• Add a final sentence on next steps or policy implications.

Reviewer #2: I am grateful for the opportunity to review this article. The following may assist in improving the manuscript further

Regarding the spread of HPV and the increase in its complications, research on it is essential.

1- The data collection tool should be explained in the text.

2. Since parental consent is required for adolescent vaccination, it is best to include and explain parental demographic characteristics in the results and discussion.

3. Were the programs listed only offered to parents of adolescents? Include in the text.

4. Did all parents of eligible adolescents have access to cell phones, websites, podcasts, etc., and were they literate enough to use them? Include in the text.

5. Since vaccination rates have been increasing over the years of the study, it should be justified that the “open door” approach had the greatest impact and was not due to prior information and other methods. Specify how confounding variables were controlled.

**Do you want your identity to be public for this peer review?** For information about this choice, including consent withdrawal, please see our Privacy Policy

Reviewer #1: No

Reviewer #2: **Yes: ** Mahnaz Azari.PhD midwifery, Reproductive Health Promotion Research Center, Ardebil University of Medical Sciences, Ardebil, Iran.

---

## [Author Response · Author response to Decision Letter 1]

31 Jul 2025

PONE-D-25-25832

Impact of Health Promotion Strategies on HPV Vaccination Uptake: A Descriptive Epidemiological Study (2019–2024)

PLOS ONE

Editor’s comments

Dear Dr. Štrbac,

Thank you for submitting your manuscript to PLOS ONE. After careful consideration, we feel that it has merit but does not fully meet PLOS ONE’s publication criteria as it currently stands. Therefore, we invite you to submit a revised version of the manuscript that addresses the points raised during the review process.

I would like you to go thoroughly across the reviews and check/change your paper in accordance with the suggestions reviewers made. I really think they made an effort to improve your findings/paper, and would like you to accept and implement everything they suggest. Please explain your changes in rebuttal letter in details. I find your paper very interesting and would like to see it published but some of the comments will certainly add specific value.

We have carefully considered all their suggestions and agree with the majority of them. We hope that our responses and the revisions made throughout the manuscript will be sufficient to support its acceptance. Our responses to the journal editorial board's requests and the reviewers’ suggestions are, as in this case, highlighted in yellow.

We look forward to receiving your revised manuscript.

Kind regards,

Iskra Alexandra Nola

Academic Editor

PLOS ONE

Authors’ response: In addition to the rebuttal letter, we submitted the 'Revised Manuscript with Track Changes' and the 'Manuscript'. In addition to the corrections made and marked in the manuscript in accordance with the reviewers’ suggestions, we have also corrected typographical errors, improved the style, and shortened certain sections to enhance clarity and reduce excessive length. We have also corrected the axis labels in Figures 1 to 4 (e.g., Number of vaccinated individuals, Years, Months, Number of promotional campaigns).

Journal Requirements:

2. We note that your Data Availability Statement is currently as follows: [All relevant data are within the manuscript and its Supporting Information files]

-The values behind the means, standard deviations and other measures reported;

-The values used to build graphs;

-The points extracted from images for analysis.

Authors’ response: We have also prepared and submitted depersonalized data related to this paper.

Authors’ response: There are no ethical restrictions for this depersonalized database.

3. We notice that your supplementary tables are included in the manuscript file. Please remove them and upload them with the file type 'Supporting Information'. Please ensure that each Supporting Information file has a legend listed in the manuscript after the references list.

Authors’ response: We removed S1 Table and S2 Table from the main text and moved their titles after the reference list, while both tables were submitted as Supporting Information.

4. We are unable to open your Supporting Information file [data.zip]. Please kindly revise as necessary and re-upload.

Authors’ response: We have re-submitted the file [data.zip], which we believe can now be opened.

Authors’ response: We have done as indicated here.

Authors’ response: We have formatted the references in accordance with the instructions provided here and the journal's requirement.

Reviewers' comments:

Reviewer's Responses to Questions

Comments to the Author

1. Is the manuscript technically sound, and do the data support the conclusions?

Reviewer #1: Yes

Reviewer #2: Partly

2. Has the statistical analysis been performed appropriately and rigorously?

Reviewer #1: Yes

Reviewer #2: Yes

3. Have the authors made all data underlying the findings in their manuscript fully available?

Reviewer #1: Yes

Reviewer #2: Yes

4. Is the manuscript presented in an intelligible fashion and written in standard English?

Reviewer #1: Yes

Reviewer #2: Yes

5. Review Comments to the Author

Response to Reviewers

Reviewer #1:

Abstract

• The sentence “...showed moderate effects. Educational efforts had limited influence.” would benefit from numerical clarity (e.g., correlation coefficients or p-values) to support the claims.

Authors’ response: Thank you for these valuable suggestions. We have added the appropriate text to the Abstract of the paper.

• Replace “among adolescents” with “among adolescents aged 9–19 years” for consistency with the main text.

Authors’ response: Thank you for this suggestion. For the sake of consistency, we have replaced the categories you mentioned throughout the manuscript.

Introduction

• Clarify the sentence “...regardless of type Serbia ranks third…” by correcting the punctuation: insert a period or semicolon between “type” and “Serbia.”

Authors’ response: Thank you for the remark. We have corrected the punctuation by inserting a [period/semicolon] between 'type' and 'Serbia' for clarity.

• Provide a citation for the “Health Policy paper” that defines HPV immunization as a “quick buy.”

Authors’ response: Thank you for this suggestion. We have inserted the reference number in the appropriate section of the manuscript.

• The aim statement is too long and somewhat repetitive. It could be tightened to a single paragraph focusing on the core research questions.

Authors’ response: Thank you for this valuable suggestion. We have shortened the aim of the study in accordance with your recommendation.

• Use consistent terminology (e.g., "nanovalent HPV vaccine" vs. "nonavalent HPV vaccine").

Authors’ response: Thank you for your comment. We have reviewed the manuscript and ensured consistent use of the term “nonavalent HPV vaccine” throughout.

• Clearly define whether the campaign’s reach extended beyond Novi Sad.

Authors’ response: Thank you for this suggestion. It is possible that we did not emphasize clearly enough that this study covered only the territory of a single city in Serbia—Novi Sad, which is the second largest in the country. We will add that the study was conducted exclusively in the territory of Novi Sad in the Aims (Introduction), Methods, and Discussion sections.

Materials and Methods

• Clarify what “address databases” were used—school-registered parent contacts? Was consent implied or granted?

Authors’ response: Thank you for your questions. As suggested, we have included a more detailed explanation of this topic in the relevant part of the Materials and Methods section.

• Include the time frame of implementation for each HPS (e.g., when were TV campaigns launched?).

Authors’ response: Thank you for these suggestions. As before, we have added a more detailed explanation of this topic in the relevant part of the Materials and Methods section.

• Elaborate on the regression model's specifications—how was the time lag introduced and validated?

Authors’ response: Thank you for your valuable comment regarding the specifications of the regression model and the introduction of the time lag. To investigate the delayed effects of promotional activities on vaccination uptake, we conducted a multiple linear regression analysis using time-shifted predictor variables. Specifically, for each promotional activity (e.g., “Open Door” sessions, distribution of printed materials, educational measures), the variable represented the activity’s scope in the previous month, while the dependent variable was the number of vaccinated individuals in the current month. A time lag of one month was applied based on the assumption that the impact of promotional activities may not be immediate but can manifest with some delay, which was particularly evident in the years 2023 and 2024 (see Fig. 3). The one-month interval was chosen as it realistically reflects the time needed for individuals to respond to the campaigns and decide to vaccinate. Importantly, this choice of lag was statistically validated, as the regression model incorporating the one-month lag explained a significantly higher proportion of variance (R² = 0.409, F(6,65) = 7.495, p < 0.001) compared to a model without any time lag. We believe this approach appropriately captures the temporal dynamics between promotional efforts and vaccination uptake.

Should the reviewer consider it useful, we are willing to incorporate a concise description of this modeling approach and the justification for the chosen time lag into the relevant section of the manuscript.

• Explain rationale for selecting the Mann–Whitney U test over t-tests or nonparametric alternatives.

Authors’ response: Thank you for your comment. The Mann–Whitney U test was selected over parametric t-tests for several reasons. First, the distribution of monthly vaccination data was highly skewed, and the sample sizes per group were relatively small, which violates the normality assumption required for independent-samples t-tests. This was confirmed by the Shapiro–Wilk test, which indicated statistically significant deviations from normality in most groups. Given these violations, the Mann–Whitney U test—a robust nonparametric alternative—was deemed appropriate, as it does not assume normality or homogeneity of variances and is well-suited for comparing two independent groups with non-normally distributed data.

These clarifications have now been included in the manuscript.

• Include a brief description of assumptions checked before using ANOVA and regression.

Authors’ response: Thank you for your comment. For ANOVA, the assumptions of normality and homogeneity of variances were assessed using the Shapiro–Wilk and Levene’s tests, respectively. The results indicated that the data for boys and girls did not consistently meet the assumption of normality (p < .05), and Levene’s test showed unequal variances between groups. Despite these limitations, ANOVA was retained for exploratory comparisons between sexes, given the equal group sizes and the value of planned post hoc analysis. All results were interpreted with caution, and nonparametric alternatives were applied to support the fin

---

## [Editor Report · Decision Letter 1]

19 Aug 2025

Impact of Health Promotion Strategies on HPV Vaccination Uptake: A Descriptive Epidemiological Study (2019–2024)

PONE-D-25-25832R1

Dear Dr. Štrbac,

We’re pleased to inform you that your manuscript has been judged scientifically suitable for publication and will be formally accepted for publication once it meets all outstanding technical requirements.

Kind regards,

Iskra Alexandra Nola

Academic Editor

PLOS ONE
---

## [Editor Report · Acceptance letter]

PONE-D-25-25832R1

PLOS ONE

Dear Dr. Štrbac,

I'm pleased to inform you that your manuscript has been deemed suitable for publication in PLOS ONE. Congratulations! Your manuscript is now being handed over to our production team.

Kind regards,

on behalf of

Dr. Iskra Alexandra Nola

Academic Editor

PLOS ONE